# Serum Levels of Selected IL-1 Family Cytokines in Patients with Morphea

**DOI:** 10.3390/jcm11216375

**Published:** 2022-10-28

**Authors:** Paulina Szczepanik-Kułak, Małgorzata Michalska-Jakubus, Małgorzata Kowal, Dorota Krasowska

**Affiliations:** Chair and Department of Dermatology, Venerology and Paediatric Dermatology, Medical University of Lublin, 20-081 Lublin, Poland

**Keywords:** morphea, localized scleroderma, IL-1 family cytokines

## Abstract

Morphea/localized scleroderma (LoS) represents an inflammatory-sclerotic skin disease, the pathogenesis of which is not fully understood. Given the important role of IL-1 family cytokines in the development and therapy of inflammatory diseases, including systemic sclerosis, we analyzed the clinical significance of serum levels of selected IL-1 family cytokines (IL-1α, IL-1β, IL-18, IL-33, IL-37 and IL-38) in LoS patients (n = 30) using the standardized disease assessment tools and comparison to healthy controls (n = 28). We also compared the pre- and post-treatment concentrations, i.e., before and after systemic (glucocorticosteroids and/or methotrexate) and/or topical (topical glucocorticosteroids and/or calcineurin inhibitors). Our findings did not reveal significant differences in baseline IL-1α, IL-1β, IL-18, IL-33, IL-37 and IL-38 levels between LoS group and HCs; however, after treatment, there were marked changes in concentrations of IL-1α and IL-33 within LoS group as well as in comparison to HCs. We also found significant negative correlations between PGA-A and IL-1α concentration as well as between mLoSSI and IL-1α after treatment. Furthermore, we showed an inverse correlation of baseline IL-1β levels with mLoSSI scores of borderline significance. We believe that IL-1α and IL-33, as well as Il-1β, may be potential mediators and targets of interest in LoS.

## 1. Introduction

Morphea, also termed localized scleroderma (LoS), is a chronic disease associated with fibrosis of the skin and possibly underlying tissues, which has the potential to progress to significant morbidity [1]. The incidence of morphea is estimated at 27 cases/1,000,000, with a marked female predominance [2,3]. There are two peaks of incidence of LoS, one between the ages of 7 and 11 and the other between the ages of 40 and 50 [4]. The pathogenesis of LoS is still not fully known [1]. The interaction of triggering factors with the individual susceptibility appears to have the major role in the disease development, resulting in the activation of both innate and acquired responses with immunoinflammatory and profibrotic actions involving the epidermis and dermis. Furthermore, in the light of evidence, including the high prevalence of an individual or family history of autoimmune disease and the presence of autoantibodies as well as certain human leukocyte antigen subtypes, morphea is believed to have an autoimmune nature [5,6].

The clinical features of LoS are heterogeneous and classified into five subtypes: limited, generalized, linear, deep and mixed [2]. In general, morphea is characterized by a consecutive stage of an active phase characterized by inflammatory or inflammatory-sclerotic, erythematous or indurative lesions with active “lilac” ring, which appeared or enlarged in the last month, followed by an inactive postinflammatory lesions with hyperpigmentation and dermal atrophy [4,7]. A variety of methods is available for the treatment of LoS [8]. For mild, superficial lesions limited to the skin, topical treatments and UV phototherapy are recommended, while for generalized, linear or deep types, systemic immunomodulatory therapies, such as glucocorticosteroids (GCS) or methotrexate (MTX), are usually introduced. In severe cases, early implementation of therapy, at the stage of inflammatory lesions, allows to avoid serious complications and disfigurement [8]. 

For the assessment of disease severity, the “localized scleroderma cutaneous assessment tool” (LoSCAT) in combination with Physician’s Global Assessment (PGA) are recommended [9]. LoSCAT includes the “modified localized scleroderma severity index” (mLoSSI) and the “localized scleroderma skin damage index” (LoSDI). mLoSSI incorporates features of disease activity or severity, such as the appearance of new skin lesions or enlargement of existing lesions in the past month, together with erythema and induration of the skin. LoSDI assesses tissue damage, reflected by atrophy of the skin and subcutaneous tissue, as well as pigmentary alterations [10]. 

To date, the search for laboratory biomarkers for LoS has not been much of a success. 

The IL-1 cytokine family comprises 11 members, including seven ligands with broad pro-inflammatory and profibrotic activities (IL-1α, IL-1β, IL-18, IL-33, IL-36α, IL-36β and IL-36γ), three receptor antagonists and one anti-inflammatory cytokine (Figure 1) [11,12,13,14]. 

IL-1 is critical to the pathogenesis of a variety of human diseases and IL-1 targeted therapies have been successfully employed to treat a range of inflammatory conditions such as rheumatoid arthritis. The role of the IL-1 family in LoS has not been extensively investigated to date [15,16,17,18,19,20,21,22,23,24,25,26,27]. Of note, involvement of the IL-1 family seems to be well established in the pathogenesis of systemic sclerosis (SSc), including development of inflammation and fibrosis in both the skin and underlying tissues, as well as internal organs [28,29,30]. 

Thus, considering the presence of some resemblances between SSc and LoS, including the similarity of in the histological pattern of skin [31], we aimed to evaluate the clinical relevance of selected serum levels of IL-1 family cytokines, IL-1α, IL-1β, IL-18, IL-33, IL-37 and IL-38 in a modest-sized single-center cohort of well-characterized and prospectively followed LoS patients and healthy controls, including comparison before and after the initiation of therapy with the use of the standardized disease assessment tools. 

## 2. Materials and Methods

### 2.1. The Study Group

The study included 30 LoS patients and 28 healthy controls (HCs), without significant difference in gender or age. All patients were hospitalized in the Department of Dermatology, Venereology, and Pediatric Dermatology Medical University of Lublin. The diagnosis of LoS was confirmed histopathologically in all cases. Informed consent was obtained from each individual before starting any procedures. The study protocol complies with the ethical guidelines of the 1975 Declaration of Helsinki. The study was approved by the local ethics committee (KE-0254/162/06/2022).

### 2.2. Clinical Assessment of Studied Individuals with LoS

First, we determined each patient’s demographics (age and sex) and age at onset of LoS diagnosis. Next, we performed a physical examination, establishing the subtype of LoS, as well as characteristics of lesions, such as activity (PGA-A, mLoSSI), damage (PGA-D, LoSDI) and body surface area (BSA). In addition, the type of LoS treatment (systemic/local) along with autoimmune co-morbidities were specified. 

The following parameters were then assessed once again after completion of the therapeutic process: PGA-A, mLoSSI, PGA-D, LoSDI, BSA.

We also evaluated the presence of ANA autoantibodies and their pattern (detected by indirect immunofluorescence using human laryngeal epidermoid carcinoma cell line type 2, Hep-2).

### 2.3. Assessment of Serum Concentrations of IL-1 Family Cytokines in LoS Patients and HCs

Peripheral blood samples (10 mL) were collected after overnight fasting—in HCs once and in LoS patients twice—before and after treatment. The blood samples were centrifuged for 15–20 min at 1000× *g* and stored at −80 °C until testing. The concentrations of following IL-1 family cytokines in serum were determined using enzyme-linked immunosorbent assay (ELISA) kits according to the manufacturer’s protocol:IL-1α: Quantikine™ELISA Human IL-1α/IL-1F1, R&D Systems, Inc., Minneapolis, MN, USA;IL-1β: Human Interleukin-1 Beta ELISA, BioVendor–Laboratorni medicina a.s., Brno, Czech Republic;IL-18: Quantikine™ELISA Human Total IL-18/IL-1F4, R&D Systems, Inc., Minneapolis, MN, USA;IL-33: Quantikine™ELISA Human IL-33, R&D Systems, Inc., Minneapolis, MN, USA;IL-37: ELISA Kit For Interleukin 1 Zeta (IL1z), Cloud-Clone Corp., Katy, Texas, USA;IL-38: ELISA Kit For Interleukin 1 Theta (IL1q), Cloud-Clone Corp., Katy, Texas, USA.

### 2.4. Statistical Methods

The statistical analysis was performed using the Statistica version 10.0 software (StatSoft Inc., Tulsa, OK, USA) for Windows.

The following statistical tests were used:Pearson’s chi-square test to compare categorical variables, including gender, between LoS patients and HCs;Mann–Whitney’s U test to compare age, cytokine concentrations between LoS patients and HCs as well as the difference in cytokine concentrations in response to systemic (immunomodulatory) treatment in LoS patients;The Wilcoxon rank-sum test to compare cytokine concentrations as well as markers of disease-targeted measures before and after treatment in patients with LoS;Spearman’s correlation coefficient to correlate markers of disease and particular interleukin levels before and after treatment in patients with LoS;McNemar test to assess alteration in disease-targeted measures following treatment.

The *p* value < 0.05 was considered statistically significant.

## 3. Results

Basic characteristics of LoS patients and HCs are shown in Table 1 and Table 2.

### 3.1. Clinical Characteristics of the Study LoS Group

The mean age of patients with LoS at the time of diagnosis was 45.4 years. The middle-aged patients predominated (median 51.5 y.o.); however, the study group also included pediatric cases (min. 10 y.o.). At baseline, we found active LoS in 28 patients. 

A considerable proportion of patients had co-occurring autoimmune diseases, most commonly vulvar lichen sclerosus (n = 8), vitiligo (n = 3) and Hashimoto’s disease (n = 2). All patients had positive ANA antibody titers, in the range of 1:80 to 1:1280, with a dominant nuclear speckled pattern (n = 25, 83.3%), less frequently nucleolar pattern (n = 5, 16.7%). In all cases, commercially available ENA/blot panels were found to be negative. Systemic therapy, in the form of GCS and MTX, was used in 13 patients (43.3%), while 17 patients (46.7%) did not require this type of therapy. All patients included in the study were treated with topical medications, topical GCS and/or calcineurin inhibitors (CI). None of the patients were on phototherapy. The data are summarized in Table 3.

### 3.2. Baseline and Post-Treatment Cytokines Levels in Patients with LoS and HCs

Baseline (pre-treatment) levels of analyzed cytokines in all of the LoS patients did not differ significantly when compared to HCs (Table 4).

After completion of the treatment, we noted an increase in IL-1α levels in our patients, and the measured concentrations were significantly higher in comparison to the HCs (*p* = 0.0440), whereas IL-33 concentrations were found to be markedly lower in LoS group after treatment in comparison to HCs (*p* = 0.0005) (Table 5). 

When comparing cytokines levels before and after treatment within LoS group, the mean and median IL-33 concentration was found to be significantly higher before therapy (*p* = 0.0110) (Table 6).

Of note, there was no significant difference in changes in cytokines levels between LoS patients on or without systemic therapy (*p* > 0.05) (Table 7).

### 3.3. Baseline and Post-Treatment Cytokines Levels in LoS Patients in Relation to Disease-Targeted Measures

In 26 of 28 patients with primarily active lesions, there was a switch to inactive phase after treatment. The decrease in the number of patients with active lesions after therapy was statistically significant (*p* < 0.0001). Mean and median disease activity (PGA-A), damage (PGA-D), mLoSSI value, LoSDI and BSA also significantly decreased after treatment (*p* < 0.05) (Table 8). 

There was no significant correlation between any of disease-targeted measures and baseline levels of analyzed cytokines (*p* > 0.05) (Table 9). 

After treatment, significant negative correlations were shown between PGA-A and IL-1α concentration (*p* = 0.0338), as well as between mLoSSI and IL-1α (*p* = 0.0338) (Table 10).

## 4. Discussion

To the best of our knowledge, this study is the first to investigate the extensive profile of IL-1 family in LoS as biomarkers of the disease and possible therapy-targeted measure.

Our findings did not reveal significant differences in baseline IL-1α, IL-1β, IL-18, IL-33, IL-37 and IL-38 levels between LoS group and HCs; however, after treatment, there were marked changes in concentrations of IL-1α and IL-33 either within LoS group as well as in comparison to HCs. Specifically, in contrast to baseline measures, after completion of therapy we obtained statistically higher values of serum IL-1α in patients with LoS than in controls, although an absolute increase in these levels within LoS group were only close to significant, probably due to a relatively small sample size. On the contrary, IL-33 concentrations significantly decreased in LoS patients under treatment and were significantly lower in post-treated LoS individuals when compared to HCs. 

In addition, our results showed some important associations of IL-1 members with certain disease-targeted measures. Particularly noteworthy, we found that IL-1α levels may have a potential strong association with disease activity and severity. In post-treated LoS patients IL-1α concentrations negatively correlated with both mLoSSi and PGA-A scores. With respect to serum levels of other analyzed cytokines, no significant association was found in terms of clinical variables in the study group; however, it is worth noting the inverse correlation of baseline IL-1β levels with mLoSSI scores with borderline significance. 

In light of these results, it seems that IL-1α and IL-33, as well as Il-1β, may be potential mediators and targets of interest in LoS. These IL-1 family members are acknowledged to play a role in inflammation and have been suggested to be involved in autoimmune diseases, including systemic lupus erythematosus and rheumatoid arthritis as well as systemic sclerosis (SSc) [28,32,33,34,35]. Moreover, both IL-1α and IL-1β have been shown to directly stimulate fibroblasts proliferation and collagen synthesis and fibroblasts are induced to secrete a range of inflammatory cytokines in response to IL-1α and IL-1β [36,37,38]. IL-1β has also been found to participate in the differentiation of Th17 cells that may play a crucial role in the development of tissue fibrosis [28,32,39]. Of note, it was demonstrated that SSc patients with coexisting LoS-like lesions exhibited overexpression of IL-1α in the epidermis of both LoS-like and typical SSc lesions [31]. On this background, our findings of lower IL-1α levels in LoS patients after treatment and their negative correlation with both mLoSSi and PGA-A scores as well as inverse association of IL-1β with mLoSSi seem controversial. However, there is a large body of literature also demonstrating an inhibitory effect of IL-1α and IL-1β on collagen synthesis [40,41]. Surprisingly, it has been shown that IL-1α increases the mRNA expression of matrix metalloproteinases (MMPs), namely, MMP-1 and MMP-3, and induces MMP-7 resulting in collagen degradation [42]. Of note, some previous studies reported increase in MMPs in LoS after treatment and their correlation with clinical improvement [43,44], but direct experimental studies on MMPs and IL-1 in LoS are lacking. 

Another possible explanation for increased post-treatment levels of IL-1α in line with improvement in disease activity and severity scores is the direct effect of applied therapies. However, data on the effects of systemic treatment on the levels of serum IL-1 family cytokines are scarce and unavailable for LoS; the paradoxical proinflammatory properties of MTX in the form of increased IL-1 as measured by secreted protein and level of gene expression have been demonstrating in vitro on human monocytic cell line U937, which is an effect that seems to be at odds with the generally anti-inflammatory activity of this drug [45]. In contrast, regarding systemic GCS, there are reports of down-regulation of IL-1α [46,47]. However, immunosuppressive and anti-inflammatory agents such as glucocorticoids may simultaneously induce increased expression of IL-1R2 (type 2 receptor for IL-1) that acts as negative regulator of the IL-1 system, modulating IL-1α availability for the signaling receptor. Interestingly, this anti-IL-1 effect seems predominantly local [32,48]. Thus, in relation to our results, it may be speculated that observed post-treatment increase in IL-1α is a kind of feedback compensatory reaction to the blocking effect of IL-1R2, making IL-1α an interesting ambiguous mediator in LoS. However, future experimental studies are warranted to elucidate this phenomenon. 

IL-33 is structurally related to IL-1β and is known to have a crucial role in immune and inflammatory reactions. Recent articles have also described IL-33 as a cytokine with emerging pro-fibrotic potential depending on targets such as IL-13, TGF-β, IFN-γ and TLR/NF-κB signaling pathways [35]. Subcutaneous injection of IL-33 in mice resulted in the development of cutaneous fibrosis, similar to that observed in patients SSc, including dermal mast cells and skin-infiltrating neutrophils through the suppression of Th1-mediated contact hypersensitivity responses [35]. Additionally, IL-33 may function as an alarmin that alerts the immune system after endothelial or epithelial cell damage during infection, physical stress, or trauma and skin trauma as well as endothelial cell damage have been proposed as crucial events in the development of LoS lesions [49,50,51]. Moreover, MTX treatment has been shown to decline skin and blood IL-33 levels in psoriatic patients [52]. 

Thus, with respect to these observations, our finding of significant decrease in serum IL-33 concentrations in LoS patients under treatment may highlight IL-33 as the possible pathogenic mediator in LoS as well as a candidate for future targeted therapies. However, since we did not found correlations with measures of disease activity or severity, further research on larger cohort is required to clarify this contradiction. 

Currently, no comparable clinical studies are available for LoS. Thus, the results are difficult to discuss. However, some essential data may be obtained from research of other diseases that may be applicable to LoS, particularly SSc that shares common inflammatory and immunologic pathways.

However, the expression levels of IL-1β and IL-1α were found significantly up-regulated in the lesional skin tissue in SSc [31,53], the serum levels of IL-1α and IL-1β are somewhat controversial. Similarl to our findings in LoS, Lin et al. [54], in line with some other studies, did not observe a significant difference in serum concentrations of IL-1α and IL-1β between SSc and HCs, although some authors have reported their elevated serum levels in SSc patients [55,56,57,58,59]. With respect to clinical variables, observations for SSc are also puzzling. In contrast to our results in LoS cohort, serum IL-1β, but not IL-1α, was positively correlated with the severity of skin involvement in SSc measured by modified Rodnan skin score (mRSS), suggesting a potential role of this cytokine in SSc fibrotic complications, but serum levels of both IL-1α and IL-1β positively correlated with carbon monoxide transfer coefficient and patients with high serum IL-1β had higher DLCO, suggesting a reduced risk of lung fibrosis and PAH [54]. 

Recently, an increasing number of studies have shown the potential role of IL-33 in SSc [28,35]. In contrast to our findings in LoS patients, serum IL-33 levels were reported to be elevated in patients with SSc compared with healthy controls and correlated with the extent of skin sclerosis as well as with the severity of pulmonary fibrosis [60]. In one recent study of IL-1α, IL-1β, IL-18 and IL-33 in a relatively modest-sized Chinese SSc cohort, only serum IL-1β and IL-33 were found to be higher in SSc in multivariable analysis; however, no clinical associations with any of these cytokines were found [59].

In conclusion, our observations may highlight the potential relevance of certain IL-1 family members, namely, IL-1α, IL-1β and, in particular, IL-33, as interesting pathogenic mediators in LoS as well as the feasibility of their use in clinical applications. However, the puzzling results obtained for IL-1α and IL-1β emphasize the need for future experimental and clinical research to clarify their role in LoS. 

We are aware of several limitations in our study, including a relatively small sample size and the restriction to a single-center population, which are likely to limit the statistical power. Moreover, due to small number of patients with inactive phase of LoS lesions at baseline, their selection as separate group for statistical analysis and comparisons was unavailable. Therefore, replication in multi-center studies with a greater number of enrolled individuals will be beneficial. 

## Figures and Tables

**Figure 1 jcm-11-06375-f001:**
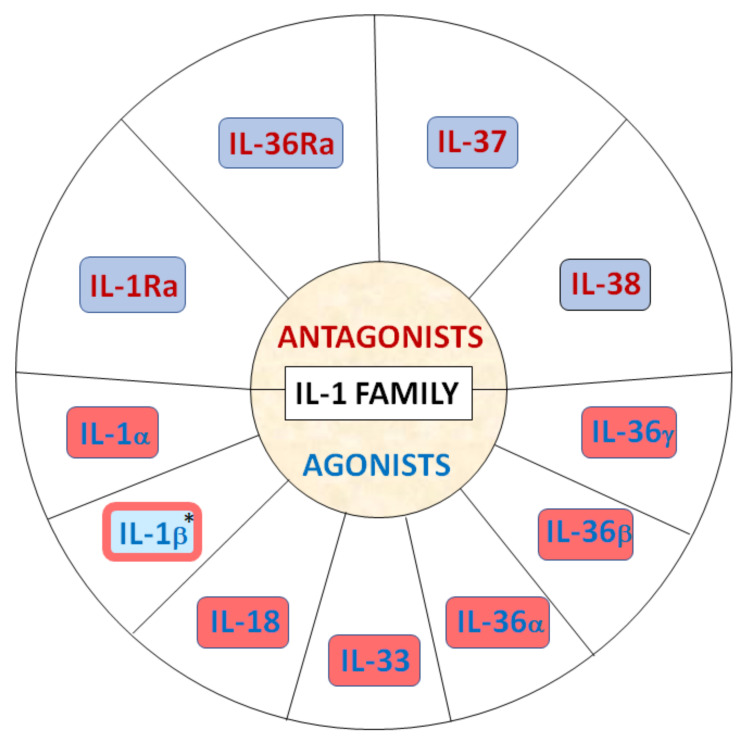
IL-1 family antagonists (highlighted in red font) and agonists (highlighted in blue font). Pro-inflammatory cytokines shown on red background and anti-inflammatory on blue background. * IL-1β may function both inflammatory and anti-inflammatory, depending on the specific receptor.

**Table 1 jcm-11-06375-t001:** Gender structure of LoS patients and HCs.

Gender	Group				χ^2^	*p*
	LoS Patients		Control			
	n	%	n	%		
Woman	26	86.7	21	75.0	1.28	0.2574
Man	4	13.3	7	25.0		
Summary	30	100.0	28	100.0		

χ^2^—Pearson’s chi-square test value.

**Table 2 jcm-11-06375-t002:** Age structure of LoS patients and HCs.

Group	n	Age [Years]					Z	*p*
		Mean	SD	Median	Min.	Max.		
LoS patients	30	53.5	18.8	61.5	11	75	1.19	0.2334
Control	28	48.5	18.7	54.5	11	75		

Z—Mann–Whitney U test value, SD—standard deviation.

**Table 3 jcm-11-06375-t003:** Clinical characteristics of patients with LoS.

Feature			
LoS onset (years)	Mean ± SD	45.4 ± 17.8	
	Median (min.–max.)	51.5 (10–73)	
		n	%
Active disease	Yes	28	93.3
	No	2	6.67
Autoimmune co-morbidities	Yes	13	43.3
	No	17	56.7
LoS type	Generalized *	25	83.3
	Linear **	5	16.7
ANA (IIF)	Positive	30	100.0
ANA pattern (IIF)	Speckled	25	83.3
	Nucleolar	5	16.7
Systemic treatment	Yes	13	43.3
	No	17	56.7
Topical treatment	Yes	30	100.0
	No	0	0.0

SD—standard deviation; IIF—indirect immunofluorescence; * four or more foci of skin sclerosis with a diameter of over 3 cm that localize in at least two anatomical areas; ** linear lesion confined to one body area.

**Table 4 jcm-11-06375-t004:** Comparison of baseline (before systemic and topical or only topical treatment) cytokines concentrations in LoS patients and HCs.

Cytokine	Group	n	Mean	SD	Median	Min.	Max.	Z	*p*
IL-1α	LoS	30	0.57	0.23	0.53	0.21	1.10	1.47	0.1417
	Control	28	0.51	0.29	0.48	0.21	1.83		
IL-1β	LoS	30	0.74	2.52	0.00	0.00	12.53	0.22	0.8282
	Control	28	2.31	9.76	0.00	0.00	51.73		
IL-18	LoS	30	159.3	63.0	145.1	71.6	298.4	−0.97	0.3308
	Control	28	192.8	111.6	163.7	74.6	621.9		
IL-33	LoS	30	0.88	1.27	0.39	0.00	5.50	−1.37	0.1697
	Control	28	1.10	1.10	0.55	0.00	3.52		
IL-37	LoS	30	0.41	2.24	0.00	0.00	12.29	−0.67	0.5035
	Control	28	3.17	13.96	0.00	0.00	72.70		
IL-38	LoS	30	0.41	2.24	0.00	0.00	12.29	−0.67	0.5035
	Control	28	3.17	13.96	0.00	0.00	72.70		

Z—Mann–Whitney U test value, SD—standard deviation.

**Table 5 jcm-11-06375-t005:** Comparison of post-treatment (after systemic and topical or only topical treatment) cytokines concentrations in LoS patients and HCs.

Cytokine	Group	n	Mean	SD	Median	Min.	Max.	Z	*p*
IL-1α	LoS	30	0.76	0.49	0.58	0.21	2.07	2.01	**0.0440**
	Control	28	0.51	0.29	0.48	0.21	1.83		
IL-1β	LoS	30	0.14	0.50	0.00	0.00	2.72	−0.71	0.4766
	Control	28	2.31	9.76	0.00	0.00	51.73		
IL-18	LoS	30	160.9	50.1	166.0	70.8	281.2	−0.63	0.5285
	Control	28	192.8	111.6	163.7	74.6	621.9		
IL-33	LoS	30	0.39	0.69	0.02	0.00	2.32	−3.46	**0.0005**
	Control	28	1.10	1.10	0.55	0.00	3.52		
IL-37	LoS	30	1.62	8.89	0.00	0.00	48.67	−0.63	0.5297
	Control	28	3.17	13.96	0.00	0.00	72.70		
IL-38	LoS	30	1.62	8.89	0.00	0.00	48.67	−0.63	0.5297
	Control	28	3.17	13.96	0.00	0.00	72.70		

SD—standard deviation; Z—Mann–Whitney U test value; *p*—probability level; Bold indicates statistically significant, *p* < 0.05.

**Table 6 jcm-11-06375-t006:** Changes in cytokines concentrations in patients with LoS before (baseline) and after treatment.

Cytokine		n	Mean	SD	Median	Min.	Max.	Z	*p*
IL-1α	Baseline	30	0.57	0.23	0.53	0.21	1.10	1.73	0.0840
	Post-treatment	30	0.76	0.49	0.58	0.21	2.07		
IL-1β	Baseline	30	0.74	2.52	0.00	0.00	12.53	0.60	0.5509
	Post-treatment	30	0.14	0.50	0.00	0.00	2.72		
IL-18	Baseline	30	159.3	63.0	145.1	71.6	298.4	0.13	0.8936
	Post-treatment	30	160.9	50.1	166.0	70.8	281.2		
IL-33	Baseline	30	0.88	1.27	0.39	0.00	5.50	2.54	**0.0110**
	Post-treatment	30	0.39	0.69	0.02	0.00	2.32		
IL-37	Baseline	30	0.41	2.24	0.00	0.00	12.29	0.45	0.6547
	Post-treatment	30	1.62	8.89	0.00	0.00	48.67		
IL-38	Baseline	30	0.41	2.24	0.00	0.00	12.29	0.45	0.6547
	Post-treatment	30	1.62	8.89	0.00	0.00	48.67		

SD—standard deviation; Z—Wilcoxon rank-sum test value; *p*—probability level; Bold indicates statistically significant, *p* < 0.05.

**Table 7 jcm-11-06375-t007:** Comparison of changes in cytokines concentrations in LoS patients depending on receiving systemic therapy.

Cytokine	Systemic Treatment	n	Mean	SD	Median	Min.	Max.	Z	*p*
IL-1α	Yes	13	0.23	0.53	0.09	−0.65	1.16	0.31	0.7536
	No	17	0.15	0.44	0.09	−0.51	1.05		
IL-1β	Yes	13	−0.03	0.08	0.00	−0.22	0.10	−0.57	0.5700
	No	17	−1.03	3.39	0.00	−12.53	1.74		
IL-18	Yes	13	−3.22	40.04	−11.29	−68.18	89.91	−0.67	0.5031
	No	17	5.29	37.89	8.30	−71.40	60.97		
IL-33	Yes	13	−0.54	1.28	−0.18	−2.32	1.86	−0.25	0.8009
	No	17	−0.45	1.23	−0.12	−3.90	2.32		
IL-37	Yes	13	−0.95	3.41	0.00	−12.29	0.00	−1.35	0.1755
	No	17	2.86	11.81	0.00	0.00	48.67		
IL-38	Yes	13	−0.95	3.41	0.00	−12.29	0.00	−1.35	0.1755
	No	17	2.86	11.81	0.00	0.00	48.67		

SD—standard deviation; *p*—probability level; Z—Mann–Whitney U test value.

**Table 8 jcm-11-06375-t008:** Results of disease-targeted measures at baseline and after treatment (only topical or combined systemic GKS/MTX and topical).

Feature		n	Mean	SD	Median	Min.	Max.	Z	*p*
PGA-A	Baseline	15	61.00	21.48	60.00	10.00	100.00	3.41	**0.0007**
	Post-treatment	15	4.00	11.21	0.00	0.00	40.00		
PGA-D	Baseline	15	31.67	20.50	25.00	10.00	70.00	3.41	**0.0007**
	Post-treatment	15	12.33	7.53	10.00	5.00	30.00		
mLoSSI	Baseline	15	22.47	19.30	13.00	1.00	59.00	3.41	**0.0007**
	Post-treatment	15	0.80	2.24	0.00	0.00	8.00		
LoSDI	Baseline	15	7.73	5.20	8.00	0.00	18.00	2.98	**0.0029**
	Post-treatment	15	4.27	2.89	4.00	1.00	11.00		
BSA (%)	Baseline	15	8.53	7.86	6.00	2.00	33.00	3.18	**0.0015**
	Post-treatment	15	6.80	7.26	4.00	2.00	30.00		

SD—standard deviation; Z—Wilcoxon rank-sum test value; *p*—probability level; Bold indicates statistically significant, *p* < 0.05.

**Table 9 jcm-11-06375-t009:** Correlations between disease-targeted measures and levels of selected cytokines in LoS group before treatment.

Before Treatment				
Variables		n	Rs	*p*
PGA-A	IL-1α	30	0.222	0.2392
	IL-1β	30	−0.197	0.2960
	IL-18	30	0.112	0.5555
	IL-33	30	0.308	0.0979
	IL-37	30	0.249	0.1849
	IL-38	30	0.249	0.1849
PGA-D	IL-1α	30	−0.082	0.6672
	IL-1β	30	0.117	0.5375
	IL-18	30	0.044	0.8170
	IL-33	30	0.234	0.2123
	IL-37	30	0.022	0.9096
	IL-38	30	0.022	0.9096
mLoSSI	IL-1α	30	0.257	0.1703
	IL-1β	30	−0.337	0.0689
	IL-18	30	−0.070	0.7138
	IL-33	30	−0.034	0.8577
	IL-37	30	0.150	0.4277
	IL-38	30	0.150	0.4277
LoSDI	IL-1α	30	0.051	0.7898
	IL-1β	30	−0.069	0.7187
	IL-18	30	0.100	0.5989
	IL-33	30	0.181	0.3389
	IL-37	30	−0.205	0.2766
	IL-38	30	−0.205	0.2766
BSA (%)	IL-1α	30	0.191	0.3111
	IL-1β	30	0.107	0.5753
	IL-18	30	0.202	0.2839
	IL-33	30	0.228	0.2257
	IL-37	30	0.248	0.1863
	IL-38	30	0.248	0.1863

Rs—value of Spearman rank correlation coefficient.

**Table 10 jcm-11-06375-t010:** Correlations between disease-targeted measures and levels of selected cytokines in LoS group after treatment.

After Treatment				
Variables		n	Rs	*p*
PGA-A	IL-1α	30	−0.389	0.0338
	IL-1β	30	−0.171	0.3653
	IL-18	30	0.296	0.1119
	IL-33	30	0.038	0.8437
	IL-37	30	−0.050	0.7946
	IL-38	30	−0.050	0.7946
PGA-D	IL-1α	30	−0.020	0.9155
	IL-1β	30	−0.266	0.1556
	IL-18	30	−0.060	0.7533
	IL-33	30	−0.120	0.5279
	IL-37	30	−0.234	0.2141
	IL-38	30	−0.234	0.2141
mLoSSI	IL-1α	30	−0.389	**0.0338**
	IL-1β	30	−0.171	0.3653
	IL-18	30	0.296	0.1119
	IL-33	30	0.038	0.8437
	IL-37	30	−0.050	0.7946
	IL-38	30	−0.050	0.7946
LoSDI	IL-1α	30	0.130	0.4934
	IL-1β	30	−0.154	0.4175
	IL-18	30	−0.309	0.0961
	IL-33	30	−0.124	0.5142
	IL-37	30	0.228	0.2251
	IL-38	30	0.228	0.2251
BSA (%)	IL-1α	30	−0.109	0.5678
	IL-1β	30	−0.117	0.5375
	IL-18	30	−0.196	0.2986
	IL-33	30	−0.097	0.6083
	IL-37	30	0.141	0.4572
	IL-38	30	0.141	0.4572

Rs—value of Spearman rank correlation coefficient; *p*—probability level; Bold indicates statistically significant, *p* < 0.05.

## Data Availability

Data sharing not applicable.

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
