# Peer review of "Serum Levels of Selected IL-1 Family Cytokines in Patients with Morphea"

_jcm, 2022, doi:10.3390/jcm11216375_

Round 1
Reviewer 1 Report
Authors evaluated the serum levels of some IL-1 family cytokines in patients with morphea. They provided some interesting data but most of the content should be improved. I also have some specific comments as the follows.
1. In the method section, authors said” All patients were hospitalized in the Department of Dermatology, Venereology, and Pediatric Dermatology Medical University of Lublin”. I wonder the reason why the patients with morphea should be hospitalized
2. Authors should well explain the reason why they chose these cytokines as the major markers. I don’t think current explanation in the introduction part is good enough.
3. Authors measured the post-treatment levels of cytokines. However, I wonder if the treatment caused significant disease improvement. Authors should show the correlation between treatment and clinical success.
4. Authors analyzed the comparison of changes in cytokines concentrations in LoS patients depending on receiving systemic therapy. However, I wonder if the treatment medication or the dosage is the same among the treatment group.
5. How to define “ active disease”?
6. Please explain the “higher values of post-treatment levels of IL-1α in LoS group than in HCs”.
7. In the Table 6, authors said “ When compared cytokines levels before and after treatment within LoS group, the mean and median IL-33 concentration was found to be significantly higher before therapy (p=0.0110)”. Because there were 17 patients who did not receive systemic therapies, authors should separately analyze the “ systemic” and “ topical treatment” group.
8. In the Table 8, what kind of treatment was applied?
9. In the Table 10, authors said “After treatment, significant negative correlations were shown between PGA-A and IL-1α concentration ….”. I also have the same question as that for Table 6. Authors should separately analyze the “ systemic” and “ topical treatment” group.
Author Response
Dr. Emmanuel Andrès
Editor-in-Chief
Journal of Clinical Medicine
Lublin, September 20 2022
Re: [JCM] Manuscript ID: jcm- 1915091
Dear Editor,
Thank You for the insightful review of our manuscript entitled: “Serum levels of selected IL-1 family cytokines in patients with morphea”. We appreciate the detailed comments of the Reviewers and we have changed our manuscript accordingly to their suggestions, as indicated below in our point-by-point reply. We have referred to literature and reconstructed the discussion to improve the quality of our manuscript.
Please find enclosed our revised manuscript (file named “jcm-1915091”) in which it has been indicated which lines were deleted (strikethrough, red colour) or added/changed (underlined, blue colour).
We hope Reviewers and Editor will be satisfied with all our responses to the comments and the revisions for the original manuscript. We feel that the changes indicated by the Reviewers have greatly improved the manuscript and we appreciate Your consideration for publication in “The Journal of Clinical Medicine”.
Yours sincerely,
Paulina Szczepanik-Kułak, MD
Chair and Department of Dermatology, Venereology and Paediatric Dermatology
Medical University of Lublin, Poland
E-mail: vpaulinav@gmail.com
Phone No.: +48 889 987 497.
Response to Comments from REVIEWER #1
Dear Reviewer,
We would like to thank you for careful and thorough reading of this manuscript and for the thoughtful comments and constructive suggestions, which help to improve the quality of this manuscript. As below, on behalf of my co-authors, I would like to clarify some of the points raised in this review.
Comment 1:
In the method section, authors said” All patients were hospitalized in the Department of Dermatology, Venereology, and Pediatric Dermatology Medical University of Lublin”. I wonder the reason why the patients with morphea should be hospitalized
Response:
Patients were hospitalized because of making the important diagnostic procedures and consultations. These with severe form of localized scleroderma also were given steroid in pulses intravenously for some days.
Comment 2:
Authors should well explain the reason why they chose these cytokines as the major markers. I don’t think current explanation in the introduction part is good enough
Response:
We would like to thank for this comment. We considered that the determination of IL-1 family cytokines would be an interesting and innovative test in morphea, especially since the search for diagnostic markers of the disease has not yielded favorable results so far. Moreover, the IL-1 is an important family of cytokines in autoimmune diseases, especially in SSc, as we explicitly emphasized in the introductory section. We sincerely hope that after this comment, the reason for our study has become clearer.
Comment 3:
Authors measured the post-treatment levels of cytokines. However, I wonder if the treatment caused significant disease improvement. Authors should show the correlation between treatment and clinical success.
Response:
Thank you for this comment; In fact, we address this issue in the top of Section 3.3. and compare baseline and post-treatment disease-targeted measures such as PGA-A, PGA-D, mLoSSI, LoSDI and BSA with Table 8. We were able to show that mean and median disease activity (PGA-A), damage (PGA-D), mLoSSI value, LoSDI and BSA significantly decreased after treatment (p < 0.05). I hope this is satisfactory.
Comment 4:
Authors analyzed the comparison of changes in cytokines concentrations in LoS patients depending on receiving systemic therapy. However, I wonder if the treatment medication or the dosage is the same among the treatment group.
Response:
Thank You for this comment. We explain that there were no significant differences in treatment algorithm for particular systemic medications in the studied LoS group. The dosages and schedule for systemic GKS or MTX were standard according to recommendations, the duration of therapy varied depending on clinical improvement; in most patients (26/28) we used the medications until disease inactivity.
Comment 5:
How to define “ active disease”?
Response:
We appreciate this question. We described the term of active lesions in the introductory section, defining them as inflammatory or inflammatory-sclerotic, erythematous or indurative with active “lilac” ring, which appeared or enlarged in the last month. Inactive lesions, on the other hand, are characterized by hyperpigmentation and dermal atrophy. Active or inactive lesions were also defined based on disease-targeted measures, i.e. modified localized scleroderma severity index (mLoSSI) incorporates features of disease activity.
Comment 6:
Please explain the “higher values of post-treatment levels of IL-1α in LoS group than in HCs”.
Response:
Thank you for your thoughtful reading of the text of our manuscript. This implies that after combination treatment (both topical and systemic) or topical only, our patients had an increase in IL-1α levels, and the labeled levels were higher compared to the control group. We have rewritten the previous section and replaced it with an explanation that we sincerely hope would be more appropriate.
Comment 7:
In the Table 6, authors said “ When compared cytokines levels before and after treatment within LoS group, the mean and median IL-33 concentration was found to be significantly higher before therapy (p=0.0110)”. Because there were 17 patients who did not receive systemic therapies, authors should separately analyze the “ systemic” and “ topical treatment” group.
Response:
We would like to thank the Reviewer very much for this comment. Please kindly note that in the next table (Table 7) we made a comparison of changes in cytokine concentrations in LoS patients depending on receiving additional systemic therapy, as topical treatment was given to all LoS patients. We did not observe significant differences depending on only topical or combined topical/systemic therapy in analyzed cytokines levels; however, it may be due to too small number of patients in both groups (17 vs. 13) to consider them as appropriate separate groups for statistical analysis.
Comment 8:
In the Table 8, what kind of treatment was applied?
Response:
This is a table that applies to all LoS patients and includes both patients receiving combination treatment (both topical and systemic) and topical only. We have included the relevant annotation “only topical or combined systemic GKS/MTX and topical” in the parenthesis in the table description.
Comment 9:
In the Table 10, authors said “After treatment, significant negative correlations were shown between PGA-A and IL-1α concentration ….”. I also have the same question as that for Table 6. Authors should separately analyze the “ systemic” and “ topical treatment” group.)
Response:
Thank you for this comment. In fact, this would be valuable; however, since we did not observe significant differences in analyzed cytokines levels depending on only topical or combined topical/systemic therapy, the separate analysis of systemic and topical treatment group in terms of both disease-targeted measures and changes in cytokine levels would be pointless from the statistical point of view and without statistical power due to too small groups for comparison. We hope this explanation will be satisfactory for the Reviewer.

Reviewer 2 Report
Szczepanik-Kulak et al describe the analysis of the IL-1 family of cytokines in LOS. This is a detailed study investigating the levels of the IL-1 cytokines between HC and LOS patients as well as levels of the cytokines before and after GCS treatment. Although the authors found no significant changes in the cytokines betweenb HC and LOS patient they did observe changes in IL-1 alpha and IL-33 after GCS treatment.
This is an important family of cytokines in autoimmune diseases therefore this analysis of the family in LOS is important.
I have a few points regarding this manuscript
In the abstract the authors should mention they are assessing cytokine levels pre and post GCS treatment instead of just state pre and post treatment. This is important information to include in the abstract.
Section 3.2/Table 4 I assume The pre-treatment analysis is before the topical GCS treatment and not systemic treatment. The author shoudl state this in the text.
Could the authors provide a hypothesis/postulate whether IL-1 and IL-33 levels would normalise again over time after GCS treatment?
Author Response
Dr. Emmanuel Andrès
Editor-in-Chief
Journal of Clinical Medicine
Lublin, September 20 2022
Re: [JCM] Manuscript ID: jcm- 1915091
Dear Editor,
Thank You for the insightful review of our manuscript entitled: “Serum levels of selected IL-1 family cytokines in patients with morphea”. We appreciate the detailed comments of the Reviewers and we have changed our manuscript accordingly to their suggestions, as indicated below in our point-by-point reply. We have referred to literature and reconstructed the discussion to improve the quality of our manuscript.
Please find enclosed our revised manuscript (file named “jcm-1915091”) in which it has been indicated which lines were deleted (strikethrough, red colour) or added/changed (underlined, blue colour).
We hope Reviewers and Editor will be satisfied with all our responses to the comments and the revisions for the original manuscript. We feel that the changes indicated by the Reviewers have greatly improved the manuscript and we appreciate Your consideration for publication in “The Journal of Clinical Medicine”.
Yours sincerely,
Paulina Szczepanik-Kułak, MD
Chair and Department of Dermatology, Venereology and Paediatric Dermatology
Medical University of Lublin, Poland
E-mail: vpaulinav@gmail.com
Phone No.: +48 889 987 497.
Response to Comments from REVIEWER #2
Dear Reviewer,
We would like to thank you for careful and thorough reading of this manuscript and for the thoughtful comments and constructive suggestions, which help to improve the quality of this manuscript. As below, on behalf of my co-authors, I would like to clarify some of the points raised in this review.
Comment 1:
In the abstract the authors should mention they are assessing cytokine levels pre and post GCS treatment instead of just state pre and post treatment. This is important information to include in the abstract.
Response:
We greatly appreciate the Reviewer for this comment. As suggested by the Reviewer, appropriate information has been added to the abstract to include this content.
Comment 2:
Section 3.2/Table 4 I assume The pre-treatment analysis is before the topical GCS treatment and not systemic treatment. The author shoudl state this in the text.
Response:
We thank the Reviewer for drawing attention to the need to include this important information. The table refers collectively to all patients with LoS, i.e. those receiving both combination treatment: systemic treatment (glucocorticosteroids and methotrexate) and topical treatment (topical corticosteroids and/or calcineurin inhibitors), as well as topical treatment in monotherapy. We have included such information in the text below the table.
Comment 3:
Could the authors provide a hypothesis/postulate whether IL-1 and IL-33 levels would normalise again over time after GCS treatment?
Response:
We are grateful for this suggestion. Unfortunately, this is a challenging conclusion to draw, especially due to the lack of studies to which we could refer and demonstrate such a relationship. However, it does prompt us to carry out further long-term studies on patients including determination of levels of selected cytokines after a longer period of disease remission.
